# Vitamin D and Healthcare Service Utilization in Children: Insights from a Machine Learning Approach

**DOI:** 10.3390/jcm11237157

**Published:** 2022-12-01

**Authors:** Giuliana Ferrante, Salvatore Fasola, Michele Piazza, Laura Tenero, Marco Zaffanello, Stefania La Grutta, Giorgio Piacentini

**Affiliations:** 1Department of Surgery, Dentistry, Pediatrics and Gynaecology, Pediatric Division, University of Verona, 37134 Verona, Italy; 2Institute of Translational Pharmacology, National Research Council, 90146 Palermo, Italy; 3AOUI Verona—Azienda Ospedaliera Universitaria Integrata of Verona, University of Verona, 37126 Verona, Italy

**Keywords:** 25(OH)D, vitamin D deficiency, paediatrics, hospitalization, regression tree, international classification of diseases

## Abstract

Vitamin D deficiency and insufficiency is a global health issue: an association has been demonstrated between vitamin D deficiency and a myriad of acute and chronic illnesses. Data regarding vitamin D status among children hospitalized with non-critical illnesses are scanty. We aimed to: (1) identify profiles of children hospitalized due to non-critical illnesses, using vitamin D levels as the driving outcome; (2) assess the association between patient profiles and length of stay. The study included 854 patients (1–17 years old) who underwent blood tests for detecting vitamin D levels. A regression tree was used to stratify patients. The relationship between vitamin D levels and length of stay was explored using Poisson regression. The regression tree identified three subgroups. Group A (16%): African, North African, Hispanic, and Indian patients. Group B (62%): Caucasian and Asian patients hospitalized for respiratory, metabolic, ill-defined, infective, and genitourinary diseases. Group C (22%): Caucasian and Asian patients hospitalized for digestive, nervous, and musculoskeletal diseases, blood and skin diseases, and injuries. Mean serum vitamin D level (ng/mL) was 13.7 (SD = 9.4) in Group A, 20.5 (10.0) in Group B, and 26.2 (12.6) in Group C. Group B was associated with the highest BMI z-score (*p* < 0.001) and the highest frequency of preterm births (*p* = 0.041). Mean length of stay was longer in Group A than in the other groups (*p* < 0.001) and decreased significantly by 9.8% (*p* = 0.024) in Group A and by 5% (*p* = 0.029) in Group B per 10 ng/mL increase in vitamin D level. We identified three subgroups of hospitalized children, defined according to ethnicity and discharge diagnosis, and characterized by increasing vitamin D levels. Vitamin D levels were associated with length of hospitalization.

## 1. Introduction

Vitamin D deficiency is a common condition worldwide at any age and plausible risk factors include non-Caucasian ethnicity, age, adiposity, low vitamin D intake, poor sun exposure, and sedentary behaviours [1]. 25-hydroxyvitamin D (25(OH)D) is the major circulating vitamin D metabolite, which dosage is generally used to evaluate the individual vitamin D status. According to the literature, different cut-offs points can be used for the definition of vitamin D status based on circulating levels of 25(OH)D: 25(OH)D < 10 ng/mL (severe deficiency), 25(OH)D < 20 ng/mL (deficiency), 20–29 ng/mL (insufficiency), and ≥30 ng/mL (sufficiency) [2].

A high prevalence of vitamin D deficiency has been reported in children in many countries across the world [3]. In US children, data indicate a prevalence ranging from 9 to 18%, and 51 to 61% of vitamin D deficiency and hypovitaminosis D, respectively [4,5]. Vitamin D deficiency is also common among healthy European children [6]. A recent meta-analysis conducted on all the European cohort studies and including 14,971 subjects aged 1–18 years, reported a prevalence of vitamin D deficiency ranging from 4 to 7%, 1 to 8%, and 12 to 40%, in different age groups (1–6 years, 7–14 years, and 15–18 years, respectively). Non-white subjects and those living in relatively mid-latitude countries had a higher prevalence than those living in southern countries [7]. Italian data are in line with the aforementioned ones, with a higher prevalence of vitamin D deficiency in neonates, adolescents, and overweight/obese subjects [8,9]. In a retrospective observational study conducted in 5096 outpatients residing in Northern Italy, the rate of 25(OH)D deficiency was 35% in Verona; according to age, a significant variation was found in the entire study population with a 26% rate of vitamin D deficiency in patients aged 1–15 years [10]. Additionally, in 1424 children and adolescents, hospitalized for different diseases in the Department of Paediatrics of the University of Verona, the reported prevalence of vitamin D deficiency was 44.2, 65.2, 69.2, 54.0, and 44.8% among Caucasians, Africans, North Africans, Indians, Hispanics, and Asians [11]. In Italy, vitamin D supplementation is recommended throughout the first year of life in all infants, regardless of type of feeding. Supplementation is much less common in children older than one year, being recommended only in children and adolescents with risk factors for vitamin D deficiency [2].

Vitamin D is a key determinant of bone health, playing a central role in the physiological regulation of calcium and phosphate transport and bone mineralization [12]. Typical manifestations of vitamin D deficiency are rickets in children and osteomalacia in adults [1]. Nonetheless, vitamin D deficiency has been associated with several disorders, including infectious, autoimmune, cardiovascular diseases, allergic asthma, and rhinitis [11,13,14,15,16], and evidence suggests that vitamin D deficiency may be related to the onset and severity of many chronic illnesses [11,17]. Indeed, in addition to calcium regulation within the gastrointestinal, renal, and skeletal systems, vitamin D may contribute to several organs’ function. For instance, it has been reported to play a relevant role in reducing inflammation and in improving immune function by mediating the innate and adaptive immune responses and triggering effective antimicrobial pathways against pathogens [18].

Although severe vitamin D deficiency is rare, many children endure a subclinical vitamin D deficiency state that may predispose them to cardiovascular, respiratory, and immune diseases [19]. Moreover, there is evidence that many hospitalized children have vitamin D deficiency, which has been associated with longer hospital stay and increased morbidity and mortality, especially in the intensive care setting [20,21,22]. However, there is a paucity of data regarding children hospitalized with non-critical illnesses. Moreover, little information is available about predictors of length of stay in this population. Birth weight, and gestational age have been identified in infants hospitalized due to bronchiolitis [23]. Nutritional status as well appeared to influence duration of hospital stay in children with non-critical conditions [24]. Estimates of costs to the healthcare system of longer hospital stays for children/adolescents with non-critical conditions as well as evidence on efficacy of vitamin D supplementation upon hospitalization in this population are not available; however, a previous study evaluating the potential economic impact of half-day reduction in length of stay for community-acquired-pneumonia hospitalizations in the US reported a significant impact, with estimated savings of USD 457 to USD 846 per episode or USD 500–900 million annually [25]. Notably, a Canadian study estimating the reductions in disease burden for increased 25(OH)D concentrations in a general population aged 3–79 years, found that if attained 25(OH)D concentrations > 100 nmol/L, the calculated reduction in annual economic burden of disease was 12.5 ± 6 billion dollars [26]. Therefore, we can assume that similar results may also be expected children/adolescents with non-critical conditions.

According to the International Classification of Diseases, Ninth Revision, Clinical Modification (ICD-9-CM), codes assigned to the discharge diagnoses of hospitalized children are classified into 17 disease groups. Therefore, analysing vitamin D differences among so many classes, while accounting for other potentially influent variables, may become cumbersome from a statistical point of view. In this regard, decision trees may be helpful to provide outcome-driven stratifications of patient profiles [27]. Regression trees are a special case of decision trees that are suitable for numerical outcomes such us vitamin D levels and involve the application of sequential binary splitting rules for partitioning the predictor space into simple regions, while maximizing the differences among the conditional means. Moreover, by providing a visual representation of the data partition process, regression trees are quite simple and useful for interpretation [28].

The primary aim of this study was taking advantage of regression trees to identify profiles of children admitted to a tertiary care hospital due to non-critical illnesses, using ICD-9-CM codes and patient characteristics as profiling variables, and vitamin D levels as the driving outcome. The secondary aim was to assess the possible association between these patient profiles and length of stay.

## 2. Materials and Methods

The study population consisted of children and adolescents of different ethnicities, consecutively admitted between January 2010 and December 2012 to the Paediatric Section of the Verona University Hospital (Italy). This is a reference hospital for diagnosis and treatment of non-critical paediatric diseases, with an average annual admission rate of 8.7%. Inclusion criteria were: age 1–17 years; residence in Verona for at least one summer season. Exclusion criteria were: mixed or uncertain ethnicity; conditions requiring chronic treatment with vitamin D supplementation. ICD-9-CM codes were assigned to the most specific discharge diagnosis per the version in which they matched. Age, gender, ethnicity, hospitalization season, gestational age, birth weight, and anthropometric data were collected. Body weight was measured to the nearest 0.1 kg, height was measured to the nearest 0.1 cm. The Institutional Ethics Committee of Verona (Italy) approved this study with approval number CE2117.

In the first or second day of stay, all patients underwent blood tests for detecting serum 25(OH)D levels, which were measured by a chemiluminescent assay (DiaSorin LIAISON automated immunoassay analyzer, DiaSorin, Stillwater, MN) and expressed as ng/mL. 

Subject characteristics were summarized through the mean and standard deviation (SDs) for quantitative variables and through absolute (percentage) frequencies for categorical variables.We used a regression tree to identify an optimal stratification of the hospitalized patients driven by vitamin D levels. Regression trees are a supervised machine-learning method in which the data are sequentially split (throughout the branches) using optimal binary categorizations of predictor values in order to obtain final groups (represented by the leaves) that are informative about the outcome. In this regard, regression trees are particularly suitable for investigating associations in the case of categorical predictors with several levels (such as ethnicity, hospitalization season, and disease group). To accomplish this, we used vitamin D level as the outcome, while the following variables were used as predictors: age (years), gender, BMI (Body Mass Index) (z-score for age), ethnicity (Caucasian, African, North African, Indian Hispanic, Asian), gestational age (preterm, i.e., <37 weeks, or full-term born, i.e., ≥37 weeks), birth weight (kg), hospitalization season, disease group (ICD-9 code), and length of stay (days). In order to reduce the number of potential patient subgroups, we set the complexity parameter of the tree to the value minimizing a cross-validated (10-fold) mean squared error, in such a way to reduce potential overfitting. The regression tree was built using the rpart package [29] of the R statistical software, version 4.0.2 (R Foundation for Statistical Computing, Vienna, Austria).

Regression trees may be subject to “masking” effects, i.e., only the strongest predictors may be used to build the tree, while other important but less strong predictors may not be used, especially when they are associated with the strongest ones (they may be seen as “surrogate” predictors). Therefore, in order to identify further potential associations, the distribution of all the predictors that were not involved in the tree was compared between the tree-based subgroups using the Kruskal–Wallis test for quantitative variables and the Chi-squared test for categorical variables. 

Within each group, the relationship between vitamin D levels and length of stay was explored using Poisson regression. In this analysis, statistical significance was set at *p* < 0.05.

## 3. Results

In the study period, a total of 3298 children and adolescents aged were admitted to the Paediatric Section of the Verona University Hospital.

Out of 854 potentially eligible children and adolescents, 27 were excluded due to infrequent diseases (less than 10 cases): mental disorders (ICD-9 code 290–319, 7 cases), congenital anomalies (740–759, 6 cases), neoplasms (140–239, 5 cases), cardiovascular diseases (390–459, 5 cases), perinatal conditions (760–779, 4 cases). The characteristics of the 827 patients included are summarized in Table 1. Mean age was 6.3 (SD = 4.3) years, 445 (54%) were males, 382 (46%) were females. Mean BMI z-score was 0.6 (1.7). Most of the patients were Caucasian (84%), and 9% were born preterm. Mean serum vitamin D level was 20.7 (11.2) ng/mL. The most frequent causes of hospitalization were respiratory (28%) and nutritional-metabolic (24%) diseases. Mean length of stay was 3.7 (2.2) days. 

The regression tree associated with the minimum cross-validated error consisted of three terminal nodes, defining three patient subgroups (Figure 1, Table 2). Group A (*n* = 129, 16%) included African, North African, Hispanic, and Indian patients (hospitalized for any disease). Group B (*n* = 518, 62%) included Caucasian and Asian patients hospitalized for respiratory, metabolic, ill-defined, infective, and genitourinary diseases. Group C (*n* = 180, 22%) included Caucasian and Asian patients hospitalized for digestive, nervous, and musculoskeletal diseases, blood and skin diseases, and injuries.

Mean serum vitamin D level was 13.7 (9.4) ng/mL in Group A, 20.5 (10.0) ng/mL in Group B, and 26.2 (12.6) ng/mL in Group C (Table 2, Figure 1 and Figure 2). Table 2 shows the distribution of the predictors that were not involved in the regression tree by group, for investigating potentially masked associations. Group B was associated with the highest BMI z-score (0.8, *p* < 0.001) and the highest frequency of preterm births (11%, *p* = 0.041). On average, patients in Group A were hospitalized for 1 day longer than in the other groups (4.6 vs. 3.6 and 3.4 days, *p* < 0.001) (Figure 2). No significant group differences were found in terms of age, gender, birth weight, and hospitalization season. 

In Group A, mean length of stay decreased significantly by 9.8% per 10-ng/ml increase in vitamin D level (*p* = 0.024). In Group B, mean length of stay decreased significantly by 5% per 10-ng/ml increase in vitamin D level *(p* = 0.029). No significant association was found in Group C (*p* = 0.821) (Figure 3). When including the other potential predictors reported in Table 2 in the Poisson regression models, the aforementioned effects remained substantially unchanged (−9.4% in Group A, *p* = 0.031, and -6% in Group B, *p* = 0.008). In Group B, mean length of stay decreased significantly by 8.4% per unit increase in the BMI z-score (*p* < 0.001). In Group C, mean length of stay decreased significantly by 25.5% per unit increase in birth weight (*p =* 0.001), while it increased in Summer (24.6%, *p* = 0.047), Autumn (33.4%, *p* = 0.014), and Winter (25.8%, *p* = 0.059), compared to Spring.

## 4. Discussion

The current study identified three subgroups of patients among those admitted to a tertiary care hospital due to non-critical illnesses, defined according to ethnicity and discharge diagnosis, and characterized by increasing vitamin D levels.

The application of a regression tree allowed us to identify an optimal way of categorizing admitted patients using the predictors involved in the tree (disease group categories and ethnicities) in order to obtain a simple, strong predictor of vitamin D levels (the three patient subgroups). 

Patients of non-Caucasian ethnicity included in this study were those with a vitamin D status ranging from severe deficiency to insufficiency, regardless of the reason for hospitalization. This is in agreement with findings from previous studies performed on hospitalized children in Italy [9,30] and could be reasonably ascribed to the less efficient vitamin D synthesis [31] as well as to less vitamin D bioavailability in people with dark skin pigmentation with respect to Caucasians [32]. Noteworthy, patients without dark skin pigmentation such as Asian and Caucasian ones were grouped together in the current study.

When investigating further potential predictors that were not involved in the regression tree (“surrogate” predictors), we found that Group B was associated with the highest BMI z-score and the highest frequency of preterm births. Vitamin D levels are known to be influenced by adiposity, given that adipose tissue is a site of storage for lipophilic substances [33] and there is evidence that overweight/obese children frequently show reduced serum vitamin D levels [34,35]. With regard to preterm birth, it is known to be associated with high risk for calcium-phosphorus metabolism alterations, with consequently impaired bone mineralization and possible development of osteopenia of prematurity [36]. Furthermore, prematurely born children have significantly lower vitamin D levels in blood serum than full-term born children, which persist until school age [37].

Noteworthy, we found that patients in Group A (non-Caucasian ethnicity and lower mean serum vitamin D levels) were hospitalized significantly longer than patients in the other groups, in agreement with previous studies in non-Caucasian children with vitamin D deficiency admitted for non-critical [38,39] and critical [22,40] illnesses. Moreover, within both Group A and Group B (Caucasian and Asian patients hospitalized for respiratory, metabolic, ill-defined, infectious, and genitourinary diseases), the expected length of stay decreased significantly per 10 ng/mL increase in vitamin D level, respectively, by 9.8% and 5%. Therefore, our findings may suggest the opportunity to supplement patients belonging to these two subgroups, following the recommendations related to their disease with regard to dosage and timing of administration of vitamin D [2], in order to reduce their hospital stay. Patients in Groups B and C were hospitalized for different diseases and showed a very similar length of hospitalization. With regard to other potential predictors of length of hospital stay in the study population, we found that factors other than vitamin D levels, such as BMI, birth weight, and seasonality influence the length of hospitalization in children and adolescents. Indeed, birth weight and BMI have been previously observed to be associated with length of stay in children hospitalized due to non-critical conditions [23,24], whereas the role of seasonality deserves further investigation. In particular, since BMI can be assumed as a proxy of socioeconomic status, we could argue that social disadvantage is associated with prolonged hospital length of stay, as it has been suggested by previous studies in children [41,42]. However, when including such potential predictors in our regression models, the effects related to vitamin D levels remained substantially unchanged.

Previous studies evaluating the efficacy of vitamin D supplementation in reducing hospital length of stay were mainly conducted in patients with respiratory infections and reported inconsistent findings. A double-blind placebo-controlled randomized clinical trial (RCT) showed that a single dose of oral vitamin D 100 000 IU reduced the time to resolution of severe pneumonia (Adjusted HR = 1.39; 95% CI = 1.11, 1.76) [43]. However, other RCTs showed no significant reduction in duration of hospital stay [44,45,46,47]. A recent Cochrane review of 7 RCTs involving a total of 1529 children with pneumonia (49% having severe or very severe disease) provided no benefit of adjunct vitamin D to antibiotic treatment on the duration of illness or hospitalization [48]. Notably, studies were heterogeneous with regard to population, disease severity, and nutritional intervention. It would be reasonably advantageous to supplement only patients who may benefit the most, namely the ones with severe deficiency, and therefore, interventional studies should be planned targeting this group rather than those with any deficiency in order to define vitamin D dosage, mode of delivery as well as duration of supplementation. Interestingly, it should be pointed out that even in children with a sufficient serum vitamin D level at hospital admission, a lower trend for the duration of resolution of severe pneumonia in hours (72 (IQR:44–96) vs. 88 (IQR:48–132); *p* = 0.07) and duration of hospital stay in days (4 (IQR:3–5) vs. 5 (IQR:4–7); *p* = 0.09) was observed in a vitamin D group compared with placebo [49]. Evidence from previous observational studies and RCTs suggested that there may be health economic benefits in raising the vitamin D level in adults in Western Europe [50]. Similar studies would also be of interest in the paediatric population.

The large sample size and the wide age range considered represent a strength of our study, along with the recruitment of consecutive patients admitted to a tertiary care hospital, hence patients belonging to every subspecialty participated in the study. Moreover, the study population was socioeconomically and ethnically diverse, although most of the patients were of Caucasian ethnicity. A prompt collection of blood samples soon after hospital admission (within 48 h) was ensured, in order to minimize the influence of factors contributing to the decline of serum vitamin D levels following hospitalization. Furthermore, this study benefitted from the advantages of regression trees, i.e., being helpful for investigating associations in the case of categorical predictors with many levels (such as the disease group), and for providing an “outcome-driven” stratification of the study population.

However, the study has some limitations. First, we are unable to fully generalize our results since only children admitted to one hospital setting have been considered. Second, we could not collect information on dietary habits and time spent outdoors. Another limitation is the lack of information about possible current vitamin D supplementation. In this regard, infants were excluded from the study, because they are usually supplemented with vitamin D; conversely, supplementation is much less common in Italian children older than 1 year, being recommended only in children and adolescents with risk factors for vitamin D deficiency [2]. Finally, 25(OH)D status was not assessed longitudinally during the hospital stay, nor was the effect of in-hospital vitamin D supplementation, and these would be important areas for future investigation.

## 5. Conclusions

In conclusion, we identified three subgroups of patients among those admitted to a tertiary care hospital due to non-critical illnesses, defined according to ethnicity and discharge diagnosis, and characterized by increasing vitamin D levels. We also found associations between vitamin D levels and length of hospitalization. Our findings may contribute to identify groups of children hospitalized due to non-critical illnesses who may most benefit from targeted vitamin D supplementation.

## Figures and Tables

**Figure 1 jcm-11-07157-f001:**
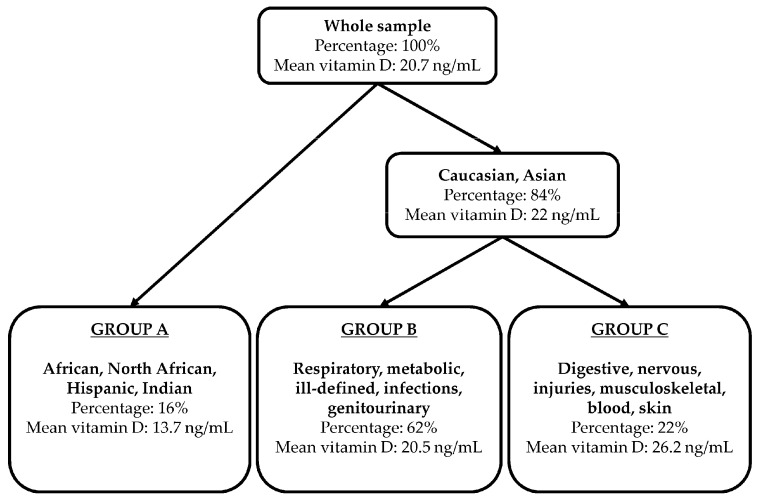
Best regression tree (according to the cross-validated accuracy) using serum vitamin D levels as the outcomes and age, gender, BMI z-score, ethnicity, gestational age, birth weight, hospitalization season, disease group, and length of stay as predictors.

**Figure 2 jcm-11-07157-f002:**
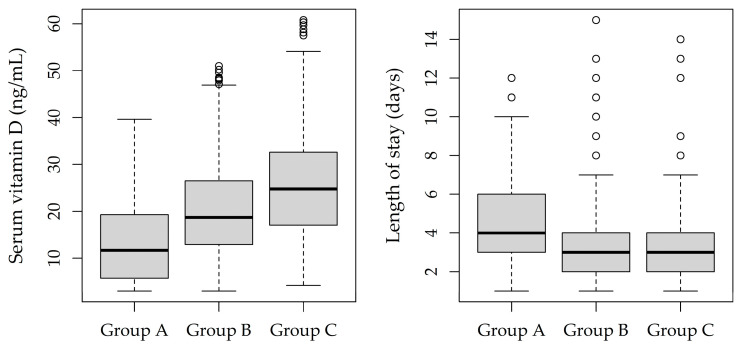
Distribution of serum vitamin D levels (left panel) and lengths of stay (right panel) by patient subgroup. Boxplots represent the median (central line), 25th–75th percentiles (box), and min-max non-outlier values (whiskers).

**Figure 3 jcm-11-07157-f003:**
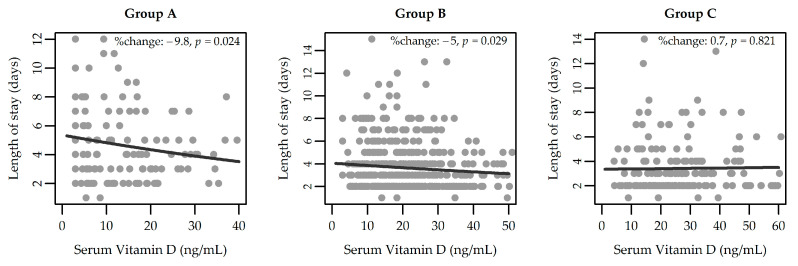
Relationship between vitamin D level and length of stay by patient subgroup. Fitted values (black lines), % change (per 10 ng/mL increase in vitamin D), and *p*-values are from Poisson regression.

**Table 1 jcm-11-07157-t001:** Patient characteristics.

Age (years), mean (SD)	6.3 (4.3)
Gender, *n* (%)	
Male	445 (54)
Female	382 (46)
BMI z-score, mean (SD)	0.6 (1.7)
Ethnicity, *n* (%)	
Caucasian	691 (84)
African	45 (5)
North African	43 (5)
Indian	27 (3)
Hispanic	14 (2)
Asian	7 (1)
Gestational age, *n* (%)	
Full-term born (≥37 weeks)	749 (91)
Preterm born (<37 weeks)	78 (9)
Birth weight (kg), mean (SD)	3.2 (0.6)
Hospitalization season, *n* (%)	
Spring	213 (26)
Summer	230 (28)
Autumn	201 (24)
Winter	183 (22)
Serum vitamin D (ng/mL), mean (SD)	20.7 (11.2)
Disease group (ICD-9 code), *n* (%)	
Respiratory diseases (460–519)	230 (28)
Nutritional-metabolic diseases (240–279)	195 (24)
Ill-defined conditions (780–799)	88 (11)
Infections (001–139)	60 (7)
Digestive diseases (520–579)	53 (6)
Nervous diseases (320–389)	43 (5)
Injuries and poisonings (800–999)	43 (5)
Genitourinary diseases (580–629)	41 (5)
Musculoskeletal diseases (710–739)	25 (3)
Blood diseases (280–289)	25 (3)
Skin diseases (680–709)	24 (3)
Length of stay (days), mean (SD)	3.7 (2.2)

**Table 2 jcm-11-07157-t002:** Distribution of the predictors that were not involved in the regression tree by group.

	Group A *n* = 129 (16%)	Group B *n* = 518 (62%)	Group C *n* = 180 (22%)	*p*-Value ^1^
Serum vitamin D (ng/mL), mean (SD)	13.7 (9.4)	20.5 (10.0)	26.2 (12.6)	<0.001
Age (years), mean (SD)	5.7 (4.2)	6.6 (4.4)	5.9 (3.9)	0.052
Gender, *n* (%)				0.753
Male	70 (54)	274 (53)	101 (56)	
Female	59 (46)	244 (47)	79 (44)	
BMI z-score, mean (SD)	0.2 (1.5)	0.8 (1.8)	0.4 (1.4)	<0.001
Gestational age, *n* (%)				0.041
Full-term born (≥37 weeks)	120 (93)	459 (89)	170 (94)	
Preterm born (<37 weeks)	9 (7)	59 (11)	10 (6)	
Birth weight (kg)	3.2 (0.6)	3.2 (0.6)	3.3 (0.5)	0.074
Hospitalization season, *n* (%)				0.205
Spring	44 (34)	119 (23)	50 (28)	
Summer	28 (22)	149 (29)	53 (29)	
Autumn	30 (23)	130 (25)	41 (23)	
Winter	27 (21)	120 (23)	36 (20)	
Length of stay (days), mean (SD)	4.6 (2.5)	3.6 (2.1)	3.4 (2.2)	<0.001

^1^ Kruskal–Wallis test for means and Chi-squared test for percentages.

## Data Availability

The data that support the findings of this study are available from the corresponding author upon reasonable request.

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
