# Peer review of "Vitamin D and Healthcare Service Utilization in Children: Insights from a Machine Learning Approach"

_jcm, 2022, doi:10.3390/jcm11237157_

Round 1

Reviewer 1 Report

This is a very interesting topic and an innovative approach. 

Introduction

It would be helpful to discuss other predictors of length of hospital stay for non-critical pediatric patients. 

It would strengthen the relevance of your research to include some estimates of costs to the healthcare system of longer hospital stays for children/adolescents with non-critical conditions, as well as mentioning costs and any evidence on efficacy of vitamin D supplementation upon hospitalization in this population.

You discuss the prevalence of vitamin D deficiency among children and adolescents in Italy, are there any more local data (Veneto region or Verona University Hospital catchment area) that can be added?

It would be helpful to include more description of decision trees in the introduction.

Methods

It would be helpful to include a description of the patient population at the Verona University Hospital Pediatric Division (average annual admission rate, characteristics of admitted population, whether this is a reference hospital, etc.).

In the introduction you note that there is a lack of data on vitamin D deficiency in children and adolescents admitted to hospital for non-critical conditions. In the methods I would have expected that you excluded critical condition admissions from eligibility for the study, is this the case? It is not clear if so. If not, you may want to reconsider this part of the introduction.

Are there any other potential predictors of length of stay that you can include in your Poisson regression models? This would make your results more convincing, even if proxy variables are used (ie. BMI for socioeconomic status, for example)

Results

I would place table 1 at the very beginning of the section.

It would be helpful to describe the eligible population (how many children/adolescents aged 1-18 were admitted to the hospital over the study period

Discussion

The first paragraphs summarize many of the results that have already been presented in the previous section. This part can be edited to avoid so much repetition.

There should be discussion of other potential predictors of length of hospital stay for this population.

There should be discussion of any literature on vitamin D supplementation in this population (or others) at the start of hospitalization and impact on length of stay.

Author Response

Reviewer 1

This is a very interesting topic and an innovative approach. 

R: We thank the Reviewer for this positive comment.

Introduction

It would be helpful to discuss other predictors of length of hospital stay for non-critical pediatric patients. 

R: We thank the Reviewer for this suggestion. A brief discussion about other predictors of length of hospital stay for non-critical pediatric patients has been included in the updated Introduction section. Additional references were provided.

It would strengthen the relevance of your research to include some estimates of costs to the healthcare system of longer hospital stays for children/adolescents with non-critical conditions, as well as mentioning costs and any evidence on efficacy of vitamin D supplementation upon hospitalization in this population.

R: Thank for this comment. A brief discussion about the suggested topic has been included in the revised Introduction section with additional references.

You discuss the prevalence of vitamin D deficiency among children and adolescents in Italy, are there any more local data (Veneto region or Verona University Hospital catchment area) that can be added?

R: We thank the Reviewer. As suggested, we included more local data in the revised Introduction section. An additional reference was provided.

It would be helpful to include more description of decision trees in the introduction.

R: Following the Reviewer suggestion, we expanded the description of decision (regression) trees in the Introduction section, also providing another reference. Thank you.

Methods

It would be helpful to include a description of the patient population at the Verona University Hospital Pediatric Division (average annual admission rate, characteristics of admitted population, whether this is a reference hospital, etc.).

R: as per reviewer’s suggest we included a description of population admitted to the Verona University Hospital in the study period. The population consisted of children and adolescents aged 1-17 years admitted for non-critical conditions. The admission rate was 8.7% per year, considering a mean value of 12500 visits in the emergency care.

In the introduction you note that there is a lack of data on vitamin D deficiency in children and adolescents admitted to hospital for non-critical conditions. In the methods I would have expected that you excluded critical condition admissions from eligibility for the study, is this the case? It is not clear if so. If not, you may want to reconsider this part of the introduction.

R: Thanks for this comment. In the updated Methods section inclusion and exclusion criteria have been included. However, we did not consider critical condition admission as an exclusion criterion given that the Paediatric Section of the Verona University Hospital only admits children affected by non-critical diseases, as we specified in the revised version of the manuscript.

Are there any other potential predictors of length of stay that you can include in your Poisson regression models? This would make your results more convincing, even if proxy variables are used (ie. BMI for socioeconomic status, for example)

R: Thank you for this comment. Following the Reviewer suggestion, in the updated Results section we specified that: “When including the other potential predictors reported in Table 2 in the Poisson regression models, the aforementioned effects remained substantially unchanged (-9.4% in Group A, p=0.031, and -6% in Group B, p=0.008). In Group B, mean length of stay decreased significantly by 8.4% per unit increase in the BMI z-score (p<0.001). In Group C, mean length of stay decreased significantly by 25.5% per unit increase in birth weight (p=0.001), while it increased in Summer (24.6%, p=0.047), Autumn (33.4%, p=0.014), and Winter (25.8%, p=0.059), compared to Spring”.

Results

I would place table 1 at the very beginning of the section.

R: Amended. Thank you.

It would be helpful to describe the eligible population (how many children/adolescents aged 1-18 were admitted to the hospital over the study period

R: We thank the Reviewer for this suggestion. In the study period were admitted to Hospital a total of 3298 children/adolescent 0-18 yo.

Discussion

The first paragraphs summarize many of the results that have already been presented in the previous section. This part can be edited to avoid so much repetition.

R: We thank the Reviewer for this suggestion. We removed some text to avoid repetition in the revised Discussion section.

There should be discussion of other potential predictors of length of hospital stay for this population.

R: Thank you. A brief discussion about other predictors of length of hospital stay for non-critical pediatric patients has been included in the updated Discussion section.

There should be discussion of any literature on vitamin D supplementation in this population (or others) at the start of hospitalization and impact on length of stay.

R: The discussion about other predictors of length of hospital stay for non-critical pediatric patients has been extended in the updated Discussion section. Additional references were provided. Thank you.

Reviewer 2 Report

Notes to the article

1.               In the introduction, the information about recommendations for vitamin D supplementation should be presented

2.               The methodology shall include information on the inclusion and exclusion criteria. Information on supplementation in the study group is also necessary. Without information about the supplementation used, it is difficult to respond to the conclusions obtained. The authors write about the importance of supplementation in the limitations.

3.               The results must be described. It is not enough to present them in figures and tables.

4.               All mean values in the article should be described with SD.

5.               In tables 1 and 2, authors shall describe exactly which values are given as n(%) and which as mean (SD)

6.               Table 2 shall describe the tests used for each analysis.

7.               In the discussion, it is necessary to refer to the differences in the diseases occurring in the subgroups, their severity, and the length of hospitalization. 

Author Response

Reviewer 2

Notes to the article

  1. In the introduction, the information about recommendations for vitamin D supplementation should be presented

R: Thanks for this suggestion. Information about recommendations for vitamin D supplementation In Italy in the updated Introduction section.

  1. The methodology shall include information on the inclusion and exclusion criteria. Information on supplementation in the study group is also necessary. Without information about the supplementation used, it is difficult to respond to the conclusions obtained. The authors write about the importance of supplementation in the limitations. 

R: We thank the Reviewer for this comment. As we stated in the Discussion, the lack of information about possible current vitamin D supplementation was acknowledged as a limitation of our study. Nonetheless, inclusion (age 1-17 years; residence in Verona for at least one summer season) and exclusion criteria (mixed or uncertain ethnicity; conditions requiring chronic treatment with vitamin D supplementation) have been included in the updated Methods section.

  1. The results must be described. It is not enough to present them in figures and tables. 

R: We apologize for this serious mistake that occurred during the production of the PDF version of the manuscript. The Results section has been included.

  1. All mean values in the article should be described with SD. 

R: Amended. Thank you.

  1. In tables 1 and 2, authors shall describe exactly which values are given as n(%) and which as mean (SD)

                        R: We amended the Tables as suggested, thank you.

  1. Table 2 shall describe the tests used for each analysis.

R: We amended Table 2 as suggested, thank you. 

  1. In the discussion, it is necessary to refer to the differences in the diseases occurring in the subgroups, their severity, and the length of hospitalization. 

R: Thank for this comment. We included a brief comment about differences in Groups B and C. However, it should be pointed out that it is not possible to define the level of disease severity. Indeed, the study population consisted only of patients admitted due to non-critical conditions. Moreover, patients in Groups B and C showed a very similar length of hospitalization, regardless of the type of disease.